# Immune Checkpoint Inhibition in Advanced Non-Clear Cell Renal Cell Carcinoma: Leveraging Success from Clear Cell Histology into New Opportunities

**DOI:** 10.3390/cancers13153652

**Published:** 2021-07-21

**Authors:** Kevin Zarrabi, Emily Walzer, Matthew Zibelman

**Affiliations:** Fox Chase Cancer Center, Department of Medical Oncology, Temple Health, Philadelphia, PA 19111, USA; Kevin.Zarrabi@tuhs.temple.edu (K.Z.); Emily.Walzer@tuhs.temple.edu (E.W.)

**Keywords:** kidney cancer, non-clear cell renal cell carcinoma, immunotherapy

## Abstract

**Simple Summary:**

Kidney cancer has many forms, which fall under the overall term, renal cell carcinoma. Clear cell renal cell carcinoma is by far the most common type, and all other renal cell carcinomas are categorized as non-clear cell renal cell carcinoma. Patients with non-clear cell renal cell carcinomas unfortunately tend to fare worse and have fewer treatment options. We are now beginning to understand non-clear cell tumors better, including what drives these tumors to grow and spread. With this improved understanding, we can design more effective therapies. The current treatments include drugs that activate the immune system to recognize and destroy cancer cells, which is termed immunotherapy. There are many active clinical trials using immunotherapy either alone or in combination with other drugs to improve the lives of patients with non-clear cell carcinomas.

**Abstract:**

Renal cell carcinoma (RCC) is a histologically heterogeneous disease with multiple subtypes. Clear cell RCC (ccRCC) represents the most common histology and has thus been easiest to study in clinical trials. Non-clear cell RCC (nccRCC) represents about 25% of RCC tumors, with fewer treatment options available, compared to ccRCC, and with poorer outcomes. Non-clear cell RCC tumors are histologically diverse, with each subtype having distinct molecular and clinical characteristics. Our understanding of nccRCC is evolving, with a gradual shift from treating nccRCC as a single entity to approaching each subtype as its own disease with unique features. Due to the scarcity of patients for study development, trials have predominantly combined all nccRCC subtypes and re-purposed drugs already approved for ccRCC, despite the decreased efficacy. We are now in the early stages of a potential paradigm shift in the treatment of nccRCC, with a rapid development of clinical studies with a focus on this subset of tumors. Investigators have launched trials focused on the molecular drivers of tumorigenesis using targeted therapies. Harboring the immunogenicity of some nccRCC subtypes, and based on promising retrospective studies, clinicians have also devised multiple trials using immune checkpoint inhibitors (ICIs), both alone or in combination with targeted therapies, for nccRCC subtypes. We highlight the promising completed and ongoing studies employing ICIs that will likely continue to improve outcomes in patients with nccRCC and propose future potential immunotherapeutic avenues.

## 1. Introduction

Renal cell carcinoma (RCC) accounts for only 3% of malignant tumors but is considered to have the highest mortality rate of the genitourinary cancers, and its annual incidence is on the rise [1,2]. RCC was previously considered a single entity, but advances in our understanding have revealed that RCC is a heterogenous disease, with various subtypes harboring unique molecular drivers and histological characteristics. The most common histology is clear cell renal cell carcinoma (ccRCC), which accounts for 70–75% of cases. The remaining 25–30% are designated under an umbrella term as non-clear cell renal cell carcinoma (nccRCC). A total of 13 subtypes of nccRCC have been identified, and each is defined by their respective anatomic and cellular origins. Some subtypes include papillary RCC (10–15%), chromophobe RCC (5%), collecting duct RCC (1%), translocation RCC (tRCC) (1%), renal medullary carcinoma (<1%), among other rare types.

Due to the fact that nccRCC tumors are histologically diverse and rare in occurrence, it has proven more challenging to conduct prospective randomized clinical trials, thus limiting our knowledge and the treatment options for these tumors [3]. In fact, many of the therapeutic strategies and systemic therapies approved for advanced nccRCC are predominantly based on data from studies on ccRCC [4]. In some cases, trials developed for nccRCC have had to stop due to a slow accrual [5]. Many of the advances in our molecular understanding of RCC, such as the identification of VHL inactivation as an oncogenic driver, is specific to ccRCC and is not applicable to nccRCC tumors. As such, these tumors are generally considered a clinical challenge for clinicians, with a paucity of data to help guide optimal therapy. This leaves a treatment landscape of therapies with a lower efficacy and with historically poor outcomes [6].

However, with the evolution of molecular genomics, we are beginning to develop a more nuanced understanding of nccRCC tumors and are recognizing many of their unique molecular alterations that drive tumorigenesis. We are witnessing the emergence of trials inclusive of patients with nccRCC and with designated nccRCC cohorts investigating molecular driver-targeted therapies. In parallel, we are concurrently seeing the emergence of immune checkpoint inhibitors (ICI) in nccRCC trial development, which was previously the cornerstone of the most recent paradigm shift witnessed in ccRCC. In light of these advances, and in the absence of prospective data supporting effective treatment strategies, ICIs have been increasingly used clinically in this setting, sometimes in combination with targeted therapies. nccRCC remains limited in approved therapies in this area, compared to ccRCC, but clinical trials are rapidly being developed to meet this urgent and unmet need.

## 2. Immunotherapy Alone in nccRCC

### 2.1. Immunotherapy with Cytokine Therapy

In the early 1990’s, cytokine-based therapy gained traction in RCC with the FDA approval of high-dose interleukin-2 (IL-2). This was after a number of clinical trials with ccRCC patients demonstrated clinical efficacy and even complete, although rare, responses. The potential of cytokine therapy for ccRCC patients prompted a series of cytokine therapy-based trials in the nccRCC population. The Programme Etude Rein Cytokines (PERCY) Quattro trial investigated medroxyprogesterone, interferon-alpha (IFN-α), IL-2, or a combination of both in 492 patients with various nccRCC histologies. No survival benefit was observed, and the treatment was associated with substantial toxicity [7]. In a separate study examining the histologic predictors of RCC response to IL-2 based therapy, patients of all histological subtypes were included, and pathologic responses were exceedingly rare in the nccRCC cohort [8]. Together, the cytokine-based therapies were not recommended for use in patients with nccRCC.

### 2.2. Immune Checkpoint Blockade: Nivolumab

With an improved understanding of the mechanisms by which neoplastic cells evade immune surveillance, we have seen the emergence of therapeutics that target immunomodulatory cell receptors. PD-1 and CTLA-4 are both cell surface glycoproteins expressed in immune cells, which serve as negative immune regulators, with a vital role in preventing autoimmunity. The ability of cancer cells to express the natural ligand to these receptors is maladaptive and may lead to unregulated cell growth inherent to tumorigenesis in some cancer types. ICIs are humanized monoclonal antibodies that selectively block the receptor to these negative regulators and enhance the activity of effector T cells, B-lymphocytes, macrophages, and natural killer cells. In doing so, the tumor microenvironment undergoes dynamic changes, allowing for an improved antitumor immune response.

The emergence of ICIs and their clinical success in ccRCC appropriately led to an evaluation of the potential immunogenicity of nccRCC tumors. Interestingly, nccRCC tumors were found to have PD-L1 cell surface expression to varying degrees, depending on the histologic subtype. Many of the evaluated intratumoral samples were also rich in tumor-infiltrating mononuclear cells. Whereas ccRCC and nccRCC tumors have little overlap in their underlying molecular characteristics, nccRCC PD-L1 expression does correlate with disease aggressiveness [9]. Collectively, these observations provide a biological rationale to apply modern immunotherapeutic strategies in nccRCC patients.

There has been a variety of retrospective hypothesis-generating work showing promise for ICI in nccRCC. A safety assessment was performed through an analysis of metastatic nccRCC cases treated with ICIs. The study identified 28 patients, most with either unclassified nccRCC or papillary RCC. In percentage terms, 39.3% of patients developed an immune-related adverse event (irAE), and 17.9% of patients developed a grade 3–4 irAE. Altogether, the patients with nccRCC treated with ICIs had an irAE profile similar to that of the ccRCC cohorts [10]. In a separate analysis, data from 43 patients (including 34 patients with nccRCC) treated with ICI monotherapy or a combination therapy (dual ICIs or ICI with VEGF-targeted therapy) were evaluated for efficacy and predictive biomarkers. The overall objective response rate (ORR) was 19%, with a 13% ORR in the ICI monotherapy subgroup. These are lower than the ORRs observed in the landmark ccRCC studies, but it is important to note that that the ORR was 31% in treatment-naïve patients. The responses were observed in patients with papillary RCC, but no responses were observed in patients with chromophobe or unclassified RCC [11]. A similar multicenter retrospective study assessed the outcomes in 40 patients with nccRCC treated with PD-1/PD-L1 inhibitors. Ten patients were treatment-naïve, and the majority received ICI monotherapy (*n* = 30), while the remaining patients received a combination of PD-1/PD-L1 and anti-VEGF or anti-CTLA-4 therapy. The ORR was 18% for the total cohort and 10% for ICI monotherapy patients, with a six-month overall survival (OS) of 81% [12].

The results of the CheckMate-025 study led to the widespread use of nivolumab in patients with refractory mRCC in clinical practice, including in patients with a non-clear cell histology, despite exclusion from the original trial population. A multicenter, retrospective analysis of the efficacy and safety of the nivolumab monotherapy in 35 patients with nccRCC using at least one dose of nivolumab was undertaken. The ORR was 20% in the total group, with all partial responses. Stable disease was noted as the best response for 29% of the patients. The study included patients with either a papillary, unclassified, chromophobe, or collecting duct histology. The ORR observed in this retrospective study was consistent with the ORR reported in the CheckMate-025 trial [13,14]. These promising findings prompted Chahoud and colleagues to perform a meta-analysis, incorporating data from 124 patients across three studies with nccRCC treated with nivolumab, assessing the ORR and disease control rates (DCR). The ORR and DCR for the total cohort were 18.6% and 53.4%, respectively. Both the ORR and DCR were consistent across cohorts, with no evidence of between-study heterogeneity or between-study inconsistency [15].

Most recently, the partial results of the multicenter phase III CheckMate-374, a study conducted to validate the safety and efficacy of nivolumab in advanced RCC, including a cohort of patients with the nccRCC histology, was released. A fixed dose of nivolumab of 240 mg every two weeks was employed. The nccRCC study cohort enrolled 44 patients who received 0 to 3 prior systemic therapies, including papillary (54.5%), chromophobe (15.9%), unclassified RCC (18.2%), translocation-associated (4.5%), collecting duct (2.3%), and medullary (2.3%) histologic subtypes. In total, 66% of patients were treatment-naïve. The median follow-up was 11.1 months, and the median duration of response (DOR) was 10.2 months. The ORR was 13.6% (95% CI 5.2–27.4). Interestingly, although the median progression-free survival (PFS) was 2.2 months, the treatment beyond progression PFS was 7.9 months. No new safety signals were identified, and there were no grade 3–5 irAEs. Altogether, these results are highly encouraging, with a clinically meaningful survival and antitumor activity and with a tolerable safety profile. Importantly, PD-L1 was neither prognostic nor predictive of treatment efficacy. Clinically meaningful efficacy results were observed, regardless of the PD-L1 expression; the mOS was 11.8 months (95% CI 8.9—NE) and 16.3 months (95% 2.9—NE) in patients with a baseline PD-L1 expression <1% and ≥1%, respectively [16]. These data are now incorporated in the European Medicines Agency’s summary of the product characteristics of nivolumab.

### 2.3. Combination Immune Checkpoint Blockade: Ipilimumab and Nivolumab

CheckMate-214 demonstrated the improved efficacy of combined ICIs in ccRCC patients, with improved response rates. It is worth noting that it included durable responses, with 53% of patients in the intention-to-treat population of the combination immunotherapy arm alive at 48 months [17]. Based on the encouraging potential in terms of durable efficacy in some ccRCC patients, some nccRCC patients elected to be treated with combination ipilimumab and nivolumab based on extrapolated data. A retrospective analysis of 18 treatment-naïve and previously treated patients, who were treated with at least one dose of combination ipilimumab and nivolumab, revealed an ORR of 33.3%, with all partial responses and a median PFS of 7.1 months. With a median of 6.8 months at follow-up from the time of the first dose (range: 0.69–12.3 months), five patients were continuing treatment, with an ongoing response. An any-grade irAE occurred in 61% of patients, and a grade ≥3 toxicity was noted in 38% of patients [18]. The response rates and survival outcomes did not match those of CheckMate-214, but this patient population was heavily pretreated before the combination ICI challenge, which further complicates any comparison.

Most recently, the results from the HCRN GU16-260-Cohort B were presented at ASCO 2021. In total, 35 treatment-naïve patients with nccRCC were treated with nivolumab monotherapy (part A), allowing for treatment with salvage combination ipilimumab and nivolumab if there was progressive disease or stable disease at the prespecified 48-week time point (part B). Of the 35 patients, 19 (54%) had a papillary, 6 (17%) had a chromophobe, and 10 (29%) had an unclassified histology. The ORR in part A was 14.3%, with an additional 45.7% of patients with stable disease. None of the respondents had progressed or died at the time of the data presentation. However, in the 16 patients enrolled in part B, the ORR was a mere 6%. The best response to ipilimumab and nivolumab was a partial response in 1 patient (6%) and stable disease in 7 patients (44%). Part B grade 3–5 irAEs were seen in 7/16 (44%) patients and included one sudden death. Correlative studies, including on the PD-L1 status, whole-exome sequencing, and RNAseq, are yet to be reported [19].

### 2.4. Immune Checkpoint Blockade: Pembrolizumab

While pembrolizumab does not have approval for use as a monotherapy in advanced RCC, it is under investigation in nccRCC. The KEYNOTE-427 was a phase II, parallel arm, open-label trial evaluating pembrolizumab monotherapy at 200 mg every three weeks in previously untreated patients with either ccRCC (cohort A) or nccRCC (cohort B). The patients received treatment for 35 cycles or until progressive disease, toxicity limitations, or study withdrawal. Amid many large prospective studies limited to patients with the ccRCC histology that are being developed, this trial serves as a model for future RCC study development, intentionally creating a designated space for patients with the nccRCC histology and demonstrating the baseline activity of the single-agent PD-pathway blockade in a nccRCC population.

The study included 165 patients with various nccRCC histologic subtypes (71% papillary, 13% chromophobe, and 16% unclassified), making it the largest interventional trial on the nccRCC histology. In total, 102 patients (61.8%) had a PD-L1 combined positive score (CPS) ≥ 1. The primary endpoint was ORR. The secondary endpoints included DCR, PFS, OS, safety, and tolerability. The ORR in the intention to treat population was 26.7% (6.7% achieving a complete response), and the median DOR was 29.0 months. Interestingly, the ORR by CPS ≥ 1 and CPS < 1 status was 35.3% and 12.1%, respectively. The mPFS was 4.2 months (95% CI, 2.9 to 5.6), and the median OS was 28.9 months (95% CI, 24.3 months was not reached). The investigators also reported that patients who experienced a reduction of 80% or greater in target lesions tended to experience a durable OS benefit (Table 1).

The safety profile was similar to that of the published reports on pembrolizumab. An irAE occurred in 32.7% patients, with grade ≥ 3 events occurring in 8.5%. The most common irAEs were hypothyroidism (15.8%), hyperthyroidism (6.7%), colitis (2.4%), and hepatitis (2.4%). Adverse event mortalities occurred in two patients (1 due to pneumonia and 1 due to cardiac arrest) [20].

### 2.5. Immune Checkpoint Blockade in Select Populations: Papillary Type

A closer examination of both the retrospective work and few prospective studies on the nccRCC histology reveals varying response rates between histological subtypes. An inherent barrier to drawing clear and convincing conclusions in nccRCC studies is that these are rare tumors, with small sample sizes, and there is a significant heterogeneity within the cohorts and study populations. Differences in the distribution of patients’ histologies and prior therapies make comparisons between studies difficult [16]. The two prospective studies in the literature, CheckMate-374 and KEYNOTE-427, are prime examples of this incongruity. KEYNOTE-427 cohort B predominantly recruited patients with the papillary histology, with far greater papillary-type patients represented than in CheckMate-374. Further, KEYNOTE-427 examines the outcomes in treatment-naïve patients, whereas the population in CheckMate-374 was heavily pretreated. As such, conclusions regarding specific histological subtypes should be drawn with caution [16].

In the absence of biology-driven trials, meta-analyses and cross-study comparisons have shown response signals to ICI in papillary-type nccRCC. Papillary RCC is the most common nccRCC histologic subtype, and naturally, it is more frequently included in the nccRCC literature, compared to the other histologies. A meta-analysis of the clinical activity of ICIs in papillary RCC showed exceedingly variable response rates, ranging from 8% to 28%, depending on the cohort [4]. The largest retrospective cohort analysis of the efficacy and safety of PD-1 inhibitors in 55 papillary type nccRCC patients showed a modest ORR of 11%. However, the same study revealed an OS of 14.6 months in the entire cohort and an OS of 11.4 months for type 1 papillary RCC, 14.6 months for type 2 papillary RCC, and 17.6 months for unclassified papillary RCC [21]. These studies hint at the clinically efficacy of PD-1 inhibition in papillary-type nccRCC. These findings were later supported by sub-analyses of the papillary subgroup from KEYNOTE-427, where the response rate was 28.8%. It should be noted that in all of the reported cohorts, both retrospective and prospective, durable responses have been documented [22].

### 2.6. Immune Checkpoint Blockade in Select Populations: Chromophobe and Translocation Subtypes

Chromophobe tumors are the second most common subtype of nccRCC and account for 5% of tumors [23]. Chromophobe RCC generally behaves in an indolent manner, and many patients are candidates for cytoreductive nephrectomy or metastasectomy in cases of oligometastatic disease. However, it has a high propensity for progression after surgery and is less likely to respond to anti-VEGF TKIs once metastatic [24,25]. Unfortunately, chromophobe tumors are considered immunologically cold, with very little tumor mutational burden, a weak PD-L1 expression, and a highly suppressive immune environment [9,26]. As such, the ICI efficacy is poor, with virtually no respondents in multiple retrospective studies looking at nivolumab or combination ipilimumab with nivolumab [13,15,18]. However, in CheckMate-374, 2 of the 7 patients with chromophobe RCC had a response, one of which had a complete response [16]. KEYNOTE-427 cohort B included 21 patients with chromophobe RCC, of which 9.5% had an ORR, with a 33.3% DCR. Together, these dampen enthusiasm for the widespread use of ICIs in patients with chromophobe RCC, but it suggests that future research focused on biomarkers for optimal patient selection or combination therapy approaches may augment ICI antitumor effects and improve outcomes.

A rare nccRCC subtype worth mentioning in the context of immunotherapy is the MiTF-family tRCC. These tumors account for 1–4% of adult RCCs and 20–40% of pediatric RCCs [27]. MiTF tRCC tumors have four distinct gene mutations: MITF, TFEC, TFEB, and TFE3, the latter two being pathognomonic for this entity. However, the tRCC histology itself has several variants, each of which can be identified by a variety of distinct and complex genetic alterations. As such, the outcomes of tRCC are highly variable, with indolent and aggressive clinical courses. It is unclear whether the variety of mutations associated with tRCC all represent molecular drivers, but the mutational profiles are believed to affect tumor morphology and clinical behavior [27]. Mutational heterogeneity aside, a recent analysis revealed that 90% of tRCC tumors express PD-L1. This prompted a retrospective review of the ICI efficacy in this tumor type. Of the 24 patients reviewed, 17 received nivolumab, 3 received ipilimumab, and 4 received ICI-based combination therapy. The response rate was 16.3%, and the disease control rate was 29.2% [27]. These findings are of particular interest, as a recent integrative clinico-genomic analysis of 152 tRCC tumors revealed an exhaustion immunophenotype distinct from ccRCC. The studied tumors also harbored neoantigens and an appreciable density of tumor-infiltrating CD8+ T cells. This integrated molecular profile supports the clinical pattern consistently observed in tRCC, where patients treated with VEGFR-TKI have worse outcomes than those treated with ICI [28].

The ongoing randomized phase II AREN1721 trial being conducted through the US cooperative groups comparing axitinib/nivolumab combination therapy versus nivolumab monotherapy in tRCC will attempt to be the first randomized trial in this rare population across children and young adults to complete and demonstrate a clear role for ICIs (NCT03595124).

## 3. Combination Therapies in nccRCC

### 3.1. Targeted Therapies

Targeted therapies, including the mammalian target of rapamycin (mTOR) inhibitors and anti-angiogenesis tyrosine kinase inhibitors (TKIs), a mainstay of treatment in ccRCC, have shown a variable efficacy, depending on the nccRCC subtype. mTOR inhibitors are a serine/threonine kinase and member of the PI3K family. mTOR activation leads to the activation of various growth factors, with pro-angiogenesis and growth signaling. The landmark ASPEN and ESPN trials have each compared sunitinib with everolimus as a first-line treatment in nccRCC. Sunitinib had a clear advantage over everolimus in the ASPEN trial, but less convincingly so in the ESPN trial. The studies did not provide definitive answers, but overall, the results paralleled that of ccRCC. The trial sample sizes were small, and the populations were heterogeneous. Moreover, the ESPN study design allowed for cross-over at progression. While they showed very modest clinical benefits, these two studies provided the evidence behind the current NCCN guideline recommendation for VEGF-directed therapy with sunitinib for non-clear cell disease [29].

### 3.2. Combination Therapy: Improved Outcomes with ICI Plus Targeted Therapy

Since the advent of immunotherapy in solid tumor oncology, the approach with combination ICI and targeted therapies has intrigued investigators. Early models during the early ICI development suggested that such combination therapies work synergistically and enhance immunomodulatory effects [30]. Anti-angiogenesis agents are now known to have pleiotropic effects. VEGF inhibitors, in particular, have been shown to increase T-lymphocyte migration into tumors, and this effect has been shown to attenuate the immunosuppressive tumor microenvironment and enhance cancer cells’ sensitivity to immune checkpoint blockade, as well as their ability to bypass mechanisms of drug resistance [31,32]. Combination therapies have become the leading therapeutic strategy in ccRCC, with multiple front-line TKI/ICI combination approvals over the past few years.

ICI in combination with targeted therapy is an area with enormous potential, with trials showing varying results in nccRCC. The first prospective trial in this setting assessed the soluble VEGF-targeted mAb bevacizumab, in combination with the anti PD-L1–targeted mAb atezolizumab, among patients with nccRCC, but it also included patients with ccRCC with sarcomatoid features. In total, 60 patients were enrolled in the trial, including 42 patients with nccRCC. The ORR was 33% for the entire cohort, 50% for patients with ccRCC with sarcomatoid features, and 26% among patients with nccRCC. The mPFS was 8.3 months, but no sub-analysis was conducted to account for the histology. Interestingly, among patients with nccRCC, the PD-L1 status was predictive of response. The ORR was 67% in PDL1-positive patients and 14% in PDL1-negative patients (*p* = 0.02). The combination was well tolerated, with only 8 patients (13%) developing a grade 3 toxicity and no treatment-related grade 4–5 toxicities noted [33]. While the small sample size and inclusion of ccRCC with sarcomatoid differentiation obscures the data interpretation for nccRCC, the study showed that the regimen was well tolerated and effective in a variety of histologies.

Cabozantinib is effective as a monotherapy in papillary RCC, providing the rationale to investigate cabozantinib in combination with ICIs. In addition to being a multi-targeted TKI, cabozantinib promotes an immune permissive environment, which is thought to enhance the response to ICIs [34]. In an elegantly designed multi-cohort basket trial in advanced solid tumors (COSMIC-021), cabozantinib in combination with atezolizumab has shown to have exciting topline results. In a similar fashion to KEYNOTE-427, investigators designated a cohort for nccRCC patients (cohort 10) pretreated with ≤1 prior to VEGFR-TKI. Cohort 10 accrued 30 patients, and the preliminary results presented at the 2020 ESMO meeting revealed an ORR of 33%. PD-L1 positivity was not associated with response. Correlative flow cytometry revealed that the treatment led to increases in cytotoxic lymphocytes, a decrease in myeloid suppressor cells, and an increased ratio of NK to monocytes in the tumor microenvironment. While the sample size is small, the responses seem to be driven by the papillary subgroup, which make up 50% of the enrolled patients and have an ORR 40%, compared to only 1 of 7 responders with the chromophobe histology, which was the second most common histology enrolled in the trial [35]. A similar study investigating the same drug combination in the salvage setting after prior ICI treatment is being evaluated in the CONTACT-03 trial in RCC, including patients with papillary or unclassified nccRCC (NCT04338269). Lastly, the results from a highly anticipated phase II study evaluating cabozantinib with nivolumab combination therapy in nccRCC patients were presented at ASCO 2021. Patients with 0 to 1 prior systemic therapy were recruited (excluding those with a prior ICI treatment) across two cohorts: cohort 1 for papillary, unclassified, and tRCC; and cohort 2 for chromophobe RCC. Cohort 1 was a single-stage design that met its primary endpoint and was expanded to produce more precise estimates of ORR (*n* = 40). Cohort 2 was a Simon two-stage design that closed early for lack of efficacy (*n* = 7). Cohort 1 patients had an ORR of 48% (95% CI 31.5–63.9), mPFS of 12.5 months (95% CI 6.3–16.4), and mOS of 28 months (95% CI 16.3–NE). No responses were seen in cohort 2 with the chromophobe histology. Altogether, the safety profile was acceptable and comparable to similar TKI/ICI combinations in the efficacy data for patients with papillary, unclassified, and tRCC, which is promising. Interestingly, correlative target exome sequencing data revealed that 5/6 patients with an NF2 mutation and 4/5 patients with an FH mutation had an objective response [36].

Finally, MET inhibition has garnered considerable attention. MET inhibition in papillary RCC is currently under investigation. MET mutations frequently occur in papillary-type RCC, and the activation of the MET pathway is associated with high-grade tumors with metastatic potential [37]. A challenge in MET-targeted therapy is the heterogeneity of the mutational aberrations. Most frequently, MET overactivity is acquired through chromosomal gain—commonly, chromosome 7 or 17 duplication. There have also been reports of spice variants specific to papillary RCC, which lead to overactivation. Lastly, gene fusions between MET and TFE3 or TFEB are seen in 15% of morphologic papillary RCC cases and are believed to be underrecognized. Given this wide array of over-activating genetic drivers, investigators are also assessing the combination of MET inhibition with ICI. An early phase, single-arm trial of combination MET inhibitor savolitinib and durvalumab was performed in papillary RCC (CALYPSO trial). The study recruited 42 patients, allowing for both treatment-naïve and previously treated patients. The results revealed an ORR of 27%, a median PFS of 4.9 months, and an OS of 12.3 months. The trial included correlatives for MET status and PD-L1 expression, but the outcomes were not enhanced in biomarker-positive patients [38].

While the trials of combination ICI and TKI to date have been small in size, their results have been promising and show response rates that are higher than either class of therapy alone in nccRCC. The trials are ongoing to further investigate the role of ICIs in combination with TKIs in nccRCC (Table 2). The enrollment has recently been completed for a highly anticipated phase II investigation of triplet therapy with cabozantinib, nivolumab, and ipilimumab in nccRCC, which is expected to finish accrual in December 2021.

## 4. Future Potential Immunotherapeutic Avenues

We are witnessing a rapid advance in our understanding of the molecular interplay driving nccRCC tumors, and this is now translating into a biology-driven trial design inclusive of, and sometimes specific to, patients with nccRCC histologic subtypes. However, an information gap remains, as there has been no ‘breakthrough’ agent or combination of agents providing high response rates with durable responses. Moreover, studies are still hampered by the slow accrual, which, in some cases, has led to the suspension of trials that are deemed unsustainable. Until recently, trials have grouped all nccRCC tumors together in order to enable rapid conclusions. We now know that this is a failing strategy, as tumors are diverse entities with their own genetic underpinnings and natural history [39]. Some have called for the development and prioritization of multicenter and potentially international trials specific to each subtype to define the best therapeutic strategies [29,39]. However, KEYNOTE-427 and COSMIC-021 both represent excellent examples in which an elegant trial design may maximize the enrollment of nccRCC patients into specific cohorts, without jeopardizing the study progress. While nccRCC tumors harbor unique mutational landscapes, they also harbor shared cytogenetic profiles, with other renal cell tumors, including ccRCC [40]. Capitalizing on redundant molecular drivers between subtypes may be an effective approach for future study development.

The somatic genomic landscape of nccRCC tumors hints at possible therapeutic targets. The virtual karyotyping of tRCC tumors identified a 17 q gain, which leads to the activation of the CTLA-4 pathway [40]. tRCC oncogenesis is partly driven by TFE3/4, which is an upstream regulator of MET. TFE4 fusion mutations lead to an upregulated MET expression. In tRCC harboring a 17 q gain and TFE3 fusion, a combination of cabozantinib with ipilimumab may yield desirable responses. This strategy applies to type 1 papillary RCC tumors as well, which has MET amplification as its oncogenic driver [41]. We have particular enthusiasm for the use of cabozantinib in combination with ICI, as it is a multitarget TKI that impacts tumor-associated macrophages, increases affected T-cells, blocks the mobilization of immunosuppressive cells, and creates an immune-permissive tumor microenvironment, with the hopes of ‘awakening’ immunologically cold tumors.

Cell metabolism represents another hypothesis-generating avenue for novel therapeutic design. Oxidative stress within the tumor microenvironment is thought to blunt the effects of immunotherapy [42]. Reactive oxygen species are used by cancer and immunosuppressive cells to create immune tolerance within tumors. Emerging evidence suggests that the modulation of tumor oxidative stress contributes to the efficacy of immunotherapy [43]. This is relevant in nccRCC, given their common somatic mutations affecting oxidative stress regulation. NRF2 is a key transcription factor that activates the endogenous antioxidant response by regulating cellular antioxidant stress [44]. Mutated NRF2 was initially related to type II papillary RCC but is now recognized as a common feature among many RCC types [45]. Aberrant NRF2 activation is as a central driver of progression and is associated with a decreased survival [37]. Direct inhibitors of NRF2 and its downstream molecules are in development, and we anticipate that these may be ideal drug candidates to be combined with ICI for synergistic effects.

Investigations into cellular metabolism revealed a shared oncogenic driver between ccRCC and some nccRCC tumors. The discovery of the VHL tumor suppressor loss in ccRCC, and the subsequent overactivity of hypoxia-inducible factor 1-alpha (HIF-1α) transcription factors, facilitated anti-angiogenesis drug development and led to the development of small molecular HIF inhibitors, which are showing tremendous promise in heavily pretreated ccRCC patients [46]. Updated results from an open label phase 2 study investigating belzutifan (MK-6482), an oral HIF-2α inhibitor, for VHL disease-associated ccRCC (NCT03401788) has revealed an impressive ORR of 49.2%, with 88.5% of patients still in the study at 20.2 months follow-up [47]. In 2016, the World Health Organization classified an entity of nccRCC based on fumarate hydratase (FH) deficiency. FH mutations lead to tumorigenesis, and 20–35% of patients with a germline FH mutation will develop FH-deficient RCC. Hereditary type II papillary RCC is also associated with germline mutations to FH. Sporadic FH mutations can also occur, although less commonly. The outcomes in FH-deficient RCC are poor [4]. FH is an enzyme of the tricarboxylic acid cycle, and aberrant activity leads to an intracellular accumulation of fumarate, which acts as an oncometabolite. In addition, fumarate is responsible for the stabilization of both HIF-1α and NRF2 [45]. The discovery that a loss of FH leads to HIF-1α overexpression is hypothesis-generating and suggests a potential antitumor activity from HIF-1α inhibitors in non-clear cell RCC with FH mutations [48]. The evaluation of HIF inhibitors in FH-deficient nccRCC is an area with potential and an approach for future trial development.

With regard to immunotherapy, the results from recent prospective studies remain immature yet are encouraging that some patients with nccRCC can benefit from ICI. This baseline clinical activity in specific histologies provides the framework for ongoing investigations as part of combination therapy with molecularly targeted drugs. Identifying tumors with inflamed microenvironments is one possible approach. PD-L1 expression is not a reliable indicator of response, but gene signatures evaluating the immunogenicity of tumors has promise. In KEYNOTE-427, RNA-sequencing–based gene expression signatures and DNA alterations were assessed. The T-cell–inflamed gene-expression profile was statistically significantly associated with pembrolizumab response in cohort A [49]. The results from the cohort B correlative studies are eagerly awaited, and positive findings may inform future patient selection for ICI trial design.

The Cancer Genome Atlas project revealed mutations in bromodomain-containing genes, such as Polybromo-1 (PBRM1), in both ccRCC and some nccRCC subtypes [37]. PBRM1 is a biomarker under active investigation in ccRCC. PRBM1 mutations are associated with an improved response to anti-angiogenesis therapy and ICI therapy in ccRCC [50]. PBRM1 is encoded on a gene locus near VHL on chromosome 3 p and is thought undergo mutation early in RCC pathogenesis [51]. Chromosome 3 loss is associated with tRCC, naturally leading to an investigation of PBRM1 as a biomarker in the metastatic MiT family, tRCC. Remarkably, of the 24 patients with tRCC treated with the ICI analyzed, two patients had long lasting responses to ICI, extending beyond six months, and both of these patients harbored a PBRM1 mutation [27,40].

## 5. Conclusions

ICIs are in the early stages of evaluation in nccRCC. In the studies reported thus far, the responses vary, depending on the histological subtype, but some meaningful antitumor activity has been demonstrated. In the absence of high-level evidence, it is unlikely that immunotherapy will gain immediate approval for use across nccRCC histologies. However, novel therapeutics targeting somatic gene mutations, tumor metabolism, and modulators of the tumor immune microenvironment are all areas of active investigation. Ultimately, pairing these therapeutic strategies with ICIs may lead to a paradigm shift in the nccRCC therapeutic landscape, as has occurred for ccRCC. Beyond the histological subtype, establishing predictive biomarkers will help guide future patient selection to optimally tailor treatment in a personalized fashion. Biology-driven trials dedicated to distinct nccRCC subsets, either histology or biomarker focused, are open for enrollment and are being rapidly developed. Trial enrollment is preferred for the treatment of metastatic nccRCC patients and should be a priority.

## Figures and Tables

**Table 1 cancers-13-03652-t001:** Completed prospective single-arm studies of immunotherapy in nccRCC.

Trial	Therapy	Sample Size, by Subtype	Prior Systemic Therapy Allowed?	Outcomes
CheckMate-374 Phase III/IV	Nivolumab	Total *n* = 44 Papillary, *n* = 24 Chromophobe, *n* = 7 Unclassified, *n* = 8 Other, *n* = 5	Yes	ORR 13.6% Papillary, 2 responses Chromophobe, 2 responses Collecting Duct, 1 response Unclassified, 1 response Median PFS 2.2 months Median OS 16.3 months
KEYNOTE-427 Cohort B Phase II	Pembrolizumab	Total *n* = 165 Papillary, *n* = 118 Chromophobe, *n* = 21 Unclassified, *n* = 26	No	ORR 26.7% Papillary—28.8% Chromophobe—9.5% Unclassified—30.8%
NCT02724878 Phase II	Bevacizumab + Atezolizumab	Total *n* = 60 Clear cell w/sarcomatoid, *n* = 18 Papillary, *n* = 12 Chromophobe, *n* = 10 Unclassified, *n* = 9 TFE3 Translocation, *n* = 5 Collecting duct, *n* = 5 Medullary, *n* = 1	Yes	ORR 33% Clear cell w/sarcomatoid—55% Papillary—25% Chromophobe—10% Unclassified—33% TFE3 Translocation—20% Collecting duct—40% Medullary—100%
CALYPSO Phase I/II	Durvalumab + Savolitinib	Total *n* = 41 * Papillary, *n* = 40 * 1 patient did not receive treatment	Yes	ORR 27% Median PFS 4.9 months Median OS 12.3 months * non-treated patient excluded from analysis
COSMIC-021 Phase Ib/II	Cabozantinib + Atezolizumab	Total *n* = 30 Papillary, *n* = 15 Chromophobe, *n* = 7 Other, *n* = 8	Yes	ORR 33% Papillary—40% Chromophobe—14% Other—60%

ORR: Objective response rate; PFS: progression-free survival; OS: overall survival. (*) 1 patient did not receive treatment.

**Table 2 cancers-13-03652-t002:** Active prospective studies for patients with nccRCC employing immunotherapy.

Trial	Disease Setting	Comparator Arm	Treatment	Study Phase	Estimated Completion
NCT02724878	Advanced nccRCC	N/A	Atezolizumab + Bevacizumab	Phase II	October 2023
NCT04704219 (KEYNOTE-B61)	Untreated advanced nccRCC	N/A	Pembrolizumab + Lenvatinib	Phase II	October 2025
NCT04267120 (LENKYN Trial)	Untreated advanced nccRCC	N/A	Pembrolizumab + Lenvatinib	Phase II	July 2024
NCT04385654	Neoadjuvant Therapy for advanced nccRCC	N/A	Toripalimab + Axitinib	Phase II	June 2022
NCT04118855	Non-metastatic Locally Advanced nccRCC	N/A		Phase II	March 2026
NCT03177239 (UNISoN)	Unresectable or metastatic nccRCC -Papillary -Chromophobe - Sarcomatoid -Xp11 Translocation	N/A	Nivolumab monotherapy, if no response, Ipi + Nivo	Phase II	December 2022
NCT04413123	Unresectable or metastatic nccRCC -Papillary -Chromophobe -Unclassified -Translocation -Collecting Duct -Renal Medullary	N/A	Ipilimumab + Nivolumab + Cabozantinib	Phase II	December 2022
NCT03075423 (SUNIFORCAST)	Untreated advanced nccRCC	Sunitinib	Ipilimuab + Nivolumab	Phase II	December 2023
NCT04644432 (INDIGO)	Untreated locally advanced or metastatic nccRCC	N/A	Study includes Pembrolizumab and Nivolumab. (Patients stratified by molecular targets and biomarker profiles)	Phase II	September 2022
NCT03274258	Treated or untreated locally advanced or metastatic nccRCC -Renal medullary	N/A	Ipilimuab + Nivolumab	Phase II	July 2022
NCT03866382	Rare Genitourinary Tumors -Chromophobe -Collecting Duct -Renal Medullary -Papillary	N/A	Ipilimumab + Nivolumab + Cabozantinib	Phase II	February 2023
NCT02721732	Rare unresectable or metastatic tumors -Renal Medullary	N/A	Pembrolizumab	Phase II	December 2021
NCT02496208	Metastatic Genitourinary Tumors -Renal Medullary -Rare Kidney Cancer Histology	N/A	Nivolumab + Cabozantinib +/− Ipilimumab	Phase I	September 2021
NCT02626130	Metastatic Kidney Cancer, including nccRCC	N/A	Tremelimumab +/− cryoablation	N/A	March 2022
NCT02819596 (CALYPSO)	Metastatic ccRCC, including Papillary RCC	N/A	Durvalumab +/− Savolitinib +/− Tremelimumab	Phase II	N/R
NCT03117309	Untreated advanced RCC of any histology	Nivolumab	Ipilimumab + Nivolumab	Phase II	September 2022
NCT03595124	Metastatic Translocation/TFE3	Nivolumab	Axitnib + Nivolumab	Phase II	June 2031
NCT04338269	Advanced RCC -Clear cell -Papillary -Chromophobe -Unclassified	Cabozantinib	Cabozantinib + Atezolizumab	Phase III	December 2024

All trial information obtained through the publicly accessible clinicaltrials.gov, 1 May 2021.

## Data Availability

No new data were created or analyzed in this study. Data sharing is not applicable to this article.

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
