# Peer review of "Immune Checkpoint Inhibition in Advanced Non-Clear Cell Renal Cell Carcinoma: Leveraging Success from Clear Cell Histology into New Opportunities"

_cancers, 2021, doi:10.3390/cancers13153652_

Round 1
Reviewer 1 Report
The results were clearly presented and could be followed easily in tables.
There are some points that I would suggest.
・In line142-143, authors demonstrated that PD-L1 was not a marker of therapeutic efficacy. Please present the data by PD-L1 expression.
・In Table1, authors reported KEYNOTE-427. They reported ORR was 24.8% and papillary 25.4%, but they also demonstrated that ORR was 26.7% in line188 and ORR of papillary RCC was 28.8%. Please clarify.
・Authors reported NCT0359124 in line262-266 and NCT04338269 in line318-321, I could not find them in table2. Please add.
・In line406, the words "open label" will appear twice. Please remove.
・Some references did not have title and journal name. (i.e. reference 10, 17)
Author Response
Thank you for the thorough and thoughtful review of the paper. Enclosed is an updated manuscript with the appropriate changes made based on the recommendations. Below, we have addressed the specific recommendations in an itemized fashion.
1) In line 142-143, authors demonstrated that PD-L1 was not a marker of therapeutic efficacy. Please present the data by PD-L1 expression.
Response: Thank you for the recommendation. The data has been included accounting for PD-L1 expression.
2) In Table 1, authors reported KEYNOTE-427. They reported ORR was 24.8% and papillary 25.4%, but they also demonstrated that ORR was 26.7% in line188 and ORR of papillary RCC was 28.8%. Please clarify
Response: Thank you for bringing this discrepancy to our attention. The data in the text as well as in Table 1 have been fixed and accounts for the published data.
3) Authors reported NCT0359124 in line 262-266 and NCT04338269 in line 318-321, I could not find them in table2. Please add.
Response: These trials are now mentioned in both the script and in Table 2.
4) In line 406, the words "open label" will appear twice. Please remove.
Response: Thank you for the correction. This error has been fixed.
5) Some references did not have title and journal name. (i.e. reference 10, 17)
Response: The references have been reviewed, and all references have been corrected. Thank you for bringing this to our attention.
Reviewer 2 Report
Immunotherapy in Advanced Non-Clear Cell Renal Cell Carcinoma: Leveraging Success from Clear Cell Histology Into New Opportunities
In this manuscript, Zarrabi and colleagues proposes a review of the current literature on the therapeutic efficiency of immune checkpoint inhibitors (ICI) in advanced non-clear cell renal cell carcinoma (nccRCC). The two main sections (i.e. 2 and 3) of this paper gives insights into the benefits and limitations of ICI alone or in combination with targeted therapies (mTOR and/or tyrosine kinase inhibitors) for nccRCC treatment.
Given the constant advances in the treatment of nccRCC, any regular update on the latest clinical research in this field is welcome. To my opinion, this is a well written manuscript which only requires minor modifications before publication, as discussed below.
Minor points
Lines 3-4: the word “Immunotherapy” in the title is misleading, as the reader expects to gain some insights into all possible immunotherapy strategies (i.e, monoclonal antibodies, active immunotherapy, adoptive T cell transfer, etc.) and not only ICI. Why are these other approaches not presented in this work? Have they proven to be inefficient in nccRCC? If so, this point should be clarified and mentioned.
Lines 67, 89: it is regrettable that the authors failed to explain what immune checkpoints are as well as their relevance as targets in cancer immunotherapy. The authors should describe the function of stimulatory/inhibitory checkpoint molecules in a few sentences.
Line 269: The authors should describe the function of mTOR and the rationale behind targeting this pathway in nccRCC treatment.
Line 107: “it is important to note”..
Author Response
We thank the reviewer for their comments, and recommendations as to how to strengthen the manuscript. All revisions have been made, and itemized below.
1) Lines 3-4: the word “Immunotherapy” in the title is misleading, as the reader expects to gain some insights into all possible immunotherapy strategies (i.e, monoclonal antibodies, active immunotherapy, adoptive T cell transfer, etc.) and not only ICI. Why are these other approaches not presented in this work? Have they proven to be inefficient in nccRCC? If so, this point should be clarified and mentioned.
Response: We fully appreciate the reviewer’s suggestion, and agree that immunotherapy is a broad term and may be misleading. Unfortunately, data on immune cellular therapies, vaccine therapies, and various other forms of immune modulation in non-clear cell renal cell carcinoma are not well described in the literature. The data on immune checkpoint inhibitors is now emerging, and reviewed in the paper. The word immunotherapy has been removed from the title and replaced with immune checkpoint inhibition.
2) Lines 67, 89: it is regrettable that the authors failed to explain what immune checkpoints are as well as their relevance as targets in cancer immunotherapy. The authors should describe the function of stimulatory/inhibitory checkpoint molecules in a few sentences.
Response: Thank you for this suggestion. We have added a paragraph reviewing the fundamental principles behind immune checkpoint inhibitors – and provide a context for the reader before exploring the data in non-clear cell renal cell carcinoma.
3) Line 269: The authors should describe the function of mTOR and the rationale behind targeting this pathway in nccRCC treatment.
Response: We have added a brief explanation of mTOR inhibitor mechanism of action to provide the rationale for their employment in renal cell carcinoma.
4) Line 107: “it is important to note”..
Response: Thank you for this correction. “It’s important to note” has now been changed to read as “it is important to note…”.
Round 2
Reviewer 1 Report
It has been modified enough according to the suggestion.